# Immune Modulation by Myeloid-Derived Suppressor Cells in Diabetic Kidney Disease

**DOI:** 10.3390/ijms232113263

**Published:** 2022-10-31

**Authors:** Ching-Chuan Hsieh, Cheng-Chih Chang, Yung-Chien Hsu, Chun-Liang Lin

**Affiliations:** 1Division of General Surgery, Chang Gung Memorial Hospital, Chiayi 261363, Taiwan; 2Kidney and Diabetic Complications Research Team (KDCRT), Chang Gung Memorial Hospital, Chiayi 261363, Taiwan; 3Division of Nephrology, Chang Gung Memorial Hospital, Chiayi 261363, Taiwan

**Keywords:** diabetic kidney disease, myeloid-derived suppressor cell, immunotherapy

## Abstract

Diabetic kidney disease (DKD) frequently leads to end-stage renal disease and other life-threatening illnesses. The dysregulation of glomerular cell types, including mesangial cells, endothelial cells, and podocytes, appears to play a vital role in the development of DKD. Myeloid-derived suppressor cells (MDSCs) exhibit immunoregulatory and anti-inflammatory properties through the depletion of L-arginine that is required by T cells, through generation of oxidative stress, interference with T-cell recruitment and viability, proliferation of regulatory T cells, and through the promotion of pro-tumorigenic functions. Under hyperglycemic conditions, mouse mesangial cells reportedly produce higher levels of fibronectin and pro-inflammatory cytokines. Moreover, the number of MDSCs is noticeably decreased, weakening inhibitory immune activities, and creating an inflammatory environment. In diabetic mice, immunotherapy with MDSCs that were induced by a combination of granulocyte-macrophage colony-stimulating factor, interleukin (IL)-1β, and IL-6, reduced kidney to body weight ratio, fibronectin expression, and fibronectin accumulation in renal glomeruli, thus ameliorating DKD. In conclusion, MDSCs exhibit anti-inflammatory activities that help improve renal fibrosis in diabetic mice. The therapeutic targeting of the proliferative or immunomodulatory pathways of MDSCs may represent an alternative immunotherapeutic strategy for DKD.

## 1. Introduction

Diabetic kidney disease (DKD), also known as diabetic nephropathy, involves the chronic loss of kidney function in patients with type 1 or 2 diabetes. DKD often leads to end-stage renal disease and other life-threatening morbidities [1,2]. DKD poses a substantial economic burden on health-care systems worldwide, and its management is thus a matter of utmost urgency [3]. Hyperglycemia or a diabetic environment triggers inflammatory [4] and profibrotic reactions [5], ultimately leading to glomerulosclerosis.

Several mechanisms mediate the progression of DKD to end-stage kidney disease; among them, glomerulosclerosis—gradual, nonstop fibrosis of the glomerulus—is a key mechanism [6]. Glomerulosclerosis results from the excessive production of extracellular matrix (ECM) proteins, and expansion of the glomerular mesangium; such expansion blocks the glomerular capillaries, and gradually destroys glomerular tissue integrity and renal function [7,8]. The most common matrix proteins include collagen types I, III, and IV, and fibronectin [9]. Furthermore, the loss of mesangial cell viability is a crucial pathological event in diabetes-mediated damage to renal tissues [10,11].

DKD is characterized by the occurrence of apoptosis-induced proteinuria, and the functional loss of podocytes, followed by a decreased glomerular filtration rate (GFR) in addition to glomerulosclerosis [12]. Although mesangial cells appear to be pivotal in mediating mesangial sclerosis in DKD, this may not entirely true. Both endothelial cells and podocytes, as well as the crosstalk among all three types of glomerular cells, appear to play vital roles in the development of diabetic glomerulosclerosis. A comprehensive approach to understanding this feature must include an analysis of all three types of glomerular cells.

## 2. Glomerular Cells Associated with DKD

### 2.1. Endothelial Cells

Glomerular endothelial cells (GEC) exhibit a specialized feature called fenestration, which enables them to accommodate large-volume filtration. The glomerular endothelium is coated with a polysaccharide-rich layer comprising the glycocalyx—a meshwork of membrane-bound glycoproteins and proteoglycans, which is called the endothelial surface layer (ESL) [13,14,15]. ESL injury is considered to facilitate microproteinuria, an early sign of DKD (Figure 1).

ESL loss has been described in patients with type 1 [16,17] or type 2 [18,19] diabetes, and the development of microproteinuria in these patients leads to further damage to the ESL, resulting in systemic vascular dysfunction [17]. The loss of the ESL or its components leads to increased glomerular permeability, which may result from enzymatic digestion, mainly heparinase, and is induced by increased levels of oxidative stress and glucose [20,21]. Furthermore, an injury to the glomerular endothelial glycocalyx may lead to the increased clearance of unmodified, negatively charged albumin, which is a sign of the early albuminuric phase of DKD [15,21,22].

Reactive oxygen species (ROS) play a direct role for GEC injury in DKD pathogenesis [23]. ROS comprise free radicals, such as superoxide (O_2_^−^) or hydroxyl (•OH), and non-radicals such as hydrogen peroxide (H_2_O_2_) [24]. The diabetic milieu triggers oxidative stress responses in GEC via several endogenous pathways, including oxidative phosphorylation in mitochondria, NADPH oxidases (NOX), cytochrome P450, xanthine oxidase (XO), and uncoupled endothelial nitric oxide synthase (NOS). NOX has been considered to contribute to the initiation and progression of DKD. The biological function of NOX enzymes is in the reduction of molecular oxygen (O_2_) to superoxide (O_2_^−^), using NADPH as an electron donor [24]. XO catalyzes the oxidation of purine substrates, xanthine and hypoxanthine, to generate H_2_O_2_ and O_2_^−^ [25,26]. Mitochondria are the energy-producing organelles in cells; via oxidative phosphorylation, they generate adenosine triphosphate (ATP). GEC approximately produce > 75% of their ATP via glycolysis, despite abundant access to oxygen [27]. ROS are byproducts from the oxidative phosphorylation reaction, and a significant portion of electrons (0.2% of the oxygen consumed) normally escape the electron transport chain as superoxide anions (O_2_^−^) [28]. Excess ROS production and accumulation result in GEC damage, including nuclear dysfunction and apoptosis.

### 2.2. Mesangial Cells

In the 1990s, a general consensus was reached regarding the primary signaling mechanisms that are involved in the stimulation of ECM protein production from mesangial cells [12]. High levels of extracellular glucose increase its uptake through the overexpression of glucose transporter 1 [29,30]. The increase in glucose metabolic flux results in the activation of a series of signaling pathways. These pathways lead to increased levels of advanced glycation end products and oxidative stress [31,32,33], which alternately induce several signaling pathways that enhance the production of ECM proteins directly, by stimulating protein kinases [34,35], extracellular signal-related kinase (ERK) pathways, and transforming growth factor (TGF)-β1 synthesis [36,37]; the pathway of TGF-β1 synthesis involves autocrine and paracrine signaling to induce the production of ECM proteins. The aforementioned TGF-β1-mediated responses appear to represent the most common mechanism underlying nephrosclerosis.

Lin et al. reported that mesangial cells impair the canonical Wnt pathway, and promote pro-apoptotic activities under hyperglycemic conditions. High glucose was found to downregulate Wnt4 and Wnt5a expressions and the subsequent nuclear translocation of beta-catenin, whereas it increased glycogen synthase kinase-3beta (GSK-3beta) and caspase-3 activities, as well as apoptosis of glomerular mesangial cells. The inhibition of Wnt signaling may be attributed to the formation of oxidative radicals. Sustained Wnt/β-catenin signaling helps sustain the responses of mesangial cells under hyperglycemic conditions [38,39]. High glucose levels stimulate Ras/Rac1 signaling to enhance superoxide production, subsequently inhibiting Wnt signaling in mesangial cells. Ras induced superoxide-activated ERK-dependent fibrosis-stimulatory factor and extracellular matrix gene transcription of mesangial cells. Pretreatments with superoxide dismutase-conjugated polyethylene glycol and diphenyleneiodonium substantially regulate superoxide production and apoptosis that is induced by high levels of glucose. Reduction in oxidative stress by scavenging superoxide may provide an alternative strategy for controlling diabetes-induced early renal injury [40].

The Akt/mTOR-mediated autophagy signaling pathway leads to mesangial cell proliferation and fibrosis in patients with DKD. Chen et al. reported that long non-coding RNAs (lncRNAs) SOX2OT downregulate in streptozotocin-induced DN mice and high glucose-induced mouse mesangial cells. Moreover, the overexpression of lncRNA SOX2OT enables the decline in autophagy, and alleviates DN-induced renal injury [41]. The downregulated expression of lncRNA nuclear-enriched abundant transcript 1 (NEAT1) reduces proliferation and fibrosis in DKD via activating the Akt/mTOR signaling pathway, which may represent a novel pathological mechanism of DKD progression [42].

### 2.3. Podocytes

Podocytes constitute the epithelial lining of the Bowman’s capsule in the kidney, and participate in the regulation of GFR. The disruption of filtration slits or destruction of podocytes may lead to proteinuria, a condition that is characterized by the loss of proteins from the blood [43]. Lin et al. demonstrated that Notch-1 signaling participates in podocyte dysregulation during the development of diabetic proteinuria. High levels of glucose induce Notch-1 signaling in human podocytes, causing vascular endothelial growth factor (VEGF) overexpression, nephrin underexpression, and apoptosis augmentation. Notably, the treatment of podocytes with recombinant VEGF leads to nephrin repression and apoptosis. After use of pharmacological modulators or specific shRNA knockdown strategies, inhibition of Notch-1 signaling significantly abrogates VEGF activation and nephrin repression in high glucose-stressed podocytes, and ameliorates proteinuria in the diabetic kidney. These results highlight the importance of the Notch-1/VEGF signaling pathway in diabetic proteinuria [44]. High levels of glucose accelerate podocyte injury, and destabilize nephrin through histone deacetylase 4 (HDAC4) and microRNA(miR)-29a levels in primary renal glomeruli from streptozotocin-induced diabetic mice. Diabetic miR-29a transgenic mice have higher nephrin levels, podocyte viability, and less glomerular fibrosis, as well as inflammation. Overexpression of miR-29a attenuates the promotion of HDAC4 signaling, nephrin ubiquitination, and urinary nephrin excretion associated with diabetes, and restores nephrin acetylation. These findings demonstrate that HDAC4 signaling and miR-29a are keys to podocyte damage in DKD [45].

The receptor activator of NF-κB (RANK) has recently been identified to be induced in podocytes in the diabetic milieu. RANK overexpression leads to enhancement of the expression of NADPH oxidase 4, and its obligate partner, P22phox, in streptozotocin-treated mice. RANK mediates the development of DKD, such as albuminuria, mesangial matrix expansion, and basement membrane thickening, probably by increasing glomerular oxidative stress and pro-inflammatory cytokine production [46]. Romero et al. reported that angiotension II (AngII) promotes the expressions of parathyroid hormone–related protein (PTHrP), TGF-β1, and p27Kip1 (a cell cycle regulatory protein), which aggravate podocyte hypertrophy under hyperglycemic conditions. These findings provide new insights into the protective effects of AngII antagonists in DKD, opening new paths for intervention [47].

## 3. MDSCs

### 3.1. Origin of MDSCs

MDSCs comprise various myeloid progenitor and precursor cells. In immunocompetent hosts, immature myeloid cells produced in the bone marrow (BM) rapidly differentiate into mature granulocytes, macrophages, or dendritic cells (DCs). Mild dysregulation of the differentiation of immature myeloid cells leads to the expansion of MDSCs under various medical conditions, such as cancer, infectious diseases, trauma, transplantation, and autoimmune disorders [48].

MDSCs express various surface markers for macrophages, monocytes, and DCs, and comprise mature myeloid cells with monocytic and granulocytic morphologies [49]. Myeloid cell colonies comprise 1–5% of MDSCs. Both in vivo and in vitro experiments have indicated that approximately 33.3% of MDSCs differentiate into mature neutrophils, macrophages, and DCs, with the help of appropriate cytokines [50,51]. In mice, MDSCs are designated by the dual expression of the myeloid lineage differentiation markers Gr1 and CD11b [52]. The BM comprises 20–30% of cells with this phenotype, but these cells account for only 2–4% of spleen cells, and lack lymph nodes in healthy mice. In humans, MDSCs are typically CD14^−^CD11b^+^: they express the common myeloid marker CD33, but not mature myeloid and lymphoid markers or human leukocyte antigen-DR isotype (HLA-DR; a major histocompatibility complex class II antigen) [53,54]. MDSCs also express CD15 and comprise 0.5% of the peripheral blood mononuclear cells (PBMCs) in healthy people [55].

The differentiation of MDSCs is influenced by two groups of factors [56,57,58]. The first group contains factors that are secreted primarily by tumor cells, and enhances the development of MDSCs by stimulating myelopoiesis, and inhibiting the proliferation of mature myeloid cells. By contrast, the second group primarily comprises inflammatory cytokines and growth factors (Table 1).

### 3.2. Tumor-Derived Factors

The administration of granulocyte-macrophage colony-stimulating factor (GM-CSF)-based vaccines in patients with metastatic melanoma noticeably increases the number of CD14^+^HLA-DR^−^/low MDSCs in the peripheral blood [59]. In female transgenic mice that express the transforming rat oncogene c-erbB-2 (HER-2/neu) and subsequently develop mammary carcinoma, MDSCs can be induced by VEGF [60]. The tumor-derived pro-inflammatory cytokine IL-1β promotes immune suppression by inducing MDSCs, thereby evading immune surveillance and enabling the proliferation and outgrowth of malignant cells [61,62]. IL-6, derived from mouse models of liver and prostate cancers, is vital in determining radiation response. Irradiation-induced IL-6 and subsequent MDSC recruitment may enhance tumor regrowth [63,64]. IL-6, secreted by human esophageal cancer cells, augments MDSC recruitment, increases reactive oxygen species (ROS) and p-STAT3 levels in MDSCs, and promotes MDSCs’ suppressive effects on T-cell proliferation [65]. In mammary carcinoma, Gr-1^+^CD11b^+^ myeloid cells lack the type II TGF-β receptor gene, and directly promote tumor metastasis. This may be explained by the increased levels of TGF-β1 in tumors with TGF-βR2 deletion, and enhanced stromal cell-derived factor-1 (SDF-1)/CXCR4 and CXCL5/CXCR2 chemokine axes [66]. IFN-γ is crucial for the suppressive effects of polymorphonuclear MDSCs; however, this does not rely on STAT1 signaling or nitric oxide (NO) production [67]. Prostaglandin E2 (PGE2) and cyclooxygenase (COX)-2 activators induce CD11b^+^Gr-1^+^ MDSCs by enhancing COX-2 expression in monocytes and blocking their differentiation into mature DCs [68,69], thus strengthening the central role of COX-2–PGE2 feedback in the induction and proliferation of MDSCs. In tumor cells that express high levels of PGE and constitutively produce COX-1 and COX-2, the expression of arginase I in MDSCs is induced [70]. Furthermore, heat shock protein 72 (Hsp72) is essential for the expansion, activation, and suppressive effects of mouse and human MDSCs through the STAT3 signaling pathway. Tumor-derived exosome-associated Hsp72 governs the suppressive effects of MDSCs by activating STAT3 in a TLR2/MyD88-dependent manner [71]. The overexpression of fms-like tyrosine kinase 3 (Flt3) ligand in tumor-bearing mice results in increased numbers of MDSCs that inhibit the antitumor activity of effector immune cells [72]. The augmentation of the suppressive effects of MDSCs by the complement component C5a results from the regulation of the levels of reactive oxygen and nitrogen species [58]. Furthermore, the administration of all-trans-retinoic acid (ATRA) remarkably inhibits the proliferation of MDSCs, and increases the levels of anti-PD-L1 antibodies in patients with cervical cancer [73]. At effective concentrations (>150 ng/mL blood), ATRA considerably decreases MDSC numbers, and improves the myeloid/lymphoid DC ratio, DC function, and the antigen-specific T-cell response in the peripheral blood of patients with metastatic renal cell carcinoma [74]. The pro-inflammatory molecule S100A9 interacts with its receptor CD33 to induce the accumulation of MDSCs in the BM of mice with myelodysplastic syndromes. S100A9/CD33 activates the immunoreceptor tyrosine-based inhibition motif to induce the secretion of the immunosuppressive cytokines IL-10 and TGF-β from immature myeloid cells [75]. The pro-inflammatory protein S100A8/A9 induces Gr-1^high^CD11b^high^F4/80^−^CD80^+^IL-4Rα^+/−^arginase^+^ MDSCs through the NF-κB signaling pathway [76]. Mice that bear tumors lacking stem cell factor (SCF) exhibit markedly reduced MDSC expansion and restored proliferative responses of tumor-infiltrating T cells. The blockage of the interaction between SCF and its receptor (c-Kit) by anti-c-Kit prevents tumor-specific T-cell anergy, regulatory T (Treg) cell development, and tumor angiogenesis [77].

### 3.3. Inflammatory Cytokines and Growth Factors

GM-CSF triggers the differentiation of mouse BM cells into immunosuppressive CD11c^−^Ly-6C^+^Ly-6GlowCD11b^+^CD31^+^ER-MP58^+^asialoGM1^+^F4/80^+^ cells in vitro [78]. The administration of recombinant G-CSF in mice leads to the accumulation of MDSCs and CD4^+^Foxp3^+^ Treg cells in peripheral lymphoid organs, which substantially prolongs the survival of skin allografts [79]. MDSCs are induced primarily by hepatic stromal cells through IL-6 signaling, and produce inhibitory enzymes to reduce T-cell immunity and promote hepatocellular carcinoma progression in the tumor microenvironment [80]. The addition of polyinosinic:polycytidylic acid to standard DC polarizing mixtures, in which DCs are generated in the presence of GM-CSF and IL-4, facilitates the accumulation of MDSCs after extended stimulation [81]. Complement component 3 secreted by hepatic stromal cells promotes the development of MDSCs, ensuring the survival of islet allografts [82].

### 3.4. Mechanisms Underlying the Suppressive Effects of MDSCs

#### 3.4.1. Depletion of Nutrients Required by T Cells

The immunosuppressive activities of MDSCs are associated with the metabolism of L-arginine. L-arginine is used as a substrate for two enzymes: inducible NO synthase (iNOS), which produces NO, and arginase I, which converts L-arginine into urea and L-ornithine. MDSCs possess high levels of both iNOS and arginase I, and exhibit direct activities of these enzymes in the suppression of T-cell functions [83]. A correlation has recently been identified between the activity of arginine and regulation of T-cell proliferation [82]. The increased expression of arginase in MDSCs results in augmented L-arginine catabolism, which exhausts the non-essential amino acid from the environment. The scarcity of L-arginine impedes T-cell proliferation through a variety of different pathways, including reducing their CD3ζ expression [84], and avoiding their upregulation of the expression of the cell cycle regulators cyclin D3 and cyclin-dependent kinase 4 (CDK4) [85]. NO suppresses T-cell activity through several different mechanisms that include the inhibition of JAK3 and STAT5 in T cells [86], the reduction in MHC class II expression [87], and the induction of T-cell apoptosis [88].

#### 3.4.2. Generation of Oxidative Stress

ROS are essential for the suppressive effects of MDSCs. The increased production of ROS is a primary feature of MDSCs in cancer [89]. The inhibition of ROS production in MDSCs isolated from mice and patients with cancer completely diminishes the suppressive effects of these cells in vitro [90]. More recently, it has emerged that peroxynitrite (ONOO^−^) is a critical element of MDSC-mediated suppression of T-cell function. Peroxynitrite is a product of a chemical reaction between NO and superoxide anion (O_2_^−^), and is one of the most powerful oxidants produced in the body. It induces the nitration and nitrosylation of the amino acids cystine, methionine, tryptophan, and tyrosine [91]. Increased levels of peroxynitrite are present at sites of MDSCs and inflammatory cell accumulation, including sites of ongoing immune reactions.

#### 3.4.3. Interference with T-Cell Recruitment and Viability

MDSCs within the tumor microenvironment present high levels of Fas-ligand (FasL) to trigger apoptosis of tumor-infiltrating lymphocytes (TILs) [92]. Polymorphonuclear (PMN)-MDSCs definitely enhance CD8^+^ T-cell apoptosis through the FasL–Fas axis, which leads to local T-cell inhibition [93]. Furthermore, MDSCs can induce T-cell suppression by engaging negative checkpoint regulators programmed cell death protein 1 (PD-1), cytotoxic T-lymphocyte-associated protein 4 (CTLA-4), and T-cell immunoglobulin and mucin domain-containing protein 3 (TIM3). Importantly, MDSCs express the PD-1 ligand (PDL1) and galectin-9, which bind PD-1 and TIM3 on TILs, respectively, restraining their antitumor immune response [94,95,96]. The expression of ADAM17 (disintegrin and metalloproteinase domain-containing protein 17) in MDSCs reduces the expression of CD62L on T cells, thus limiting T-cell recruitment at lymph nodes [97]. The regulation of CCL2 by peroxynitrite produced by MDSCs impedes the movement of effector CD8^+^ T cells toward the tumor site [98]. In addition, MDSCs express galectin-9 that binds to T-cell immunoglobulin and mucin domain-containing protein 3 on lymphocytes, and induces T-cell apoptosis [99].

#### 3.4.4. Proliferation of Treg Cells

Recently, an in vivo immunosuppressive role of MDSCs in increasing the de novo generation of Foxp3^+^ Treg cells has been reported [100]. The production of Treg cells by MDSCs requires tumor-specific T-cell stimulation, interferon (IFN)-γ, and IL-10; however, this process is independent of NO [89]. In a mouse model of lymphoma, MDSCs were demonstrated to promote Treg cells expansion through a mechanism that requires arginase, and the capture, processing, and presentation of tumor-associated antigens by MDSCs, but not TGFβ [101]. By contrast, Movahedi et al. found that the percentage of Treg cells is consistent with tumor growth, and does not relate to the kinetics of propagation of the MDSC population, suggesting that MDSCs do not participate in Treg-cell expansion [67]. The emerging evidence makes it possible that MDSCs are involved in Treg cell differentiation through the production of cytokines or direct cell–cell interactions.

#### 3.4.5. Promotion of Pro-Tumorigenic Functions

The enhancement of angiogenesis and suppression of host immunity are central to tumorigenesis. Notably, MDSCs exhibit both of the aforementioned activities, and create an environment to facilitate tumor progression. The number of MDSCs is higher in patients with cancer; this accumulation is triggered by angiogenic factors and an inflammatory environment [102]. MDSCs also produce type 2 macrophages that promote tumor progression [48]. Moreover, they penetrate tumors and enhance tumor vascularization, development, and metastasis, by regulating VEGF and protease activity in the tumor microenvironment. Boelte et al. identified a crucial role (mediated through the modulation of monocyte chemoattractant protein 1) of the regulation factor of G-protein signaling-2 in the proangiogenic activity of MDSCs in the tumor microenvironment [103]. Neutrophilic granule protein, a negative factor for tumor metastasis in MDSCs, is downregulated in metastatic conditions [104]. Together, these findings suggest that tumor invasion and metastasis are promoted through the upregulation of proteases, and the downregulation of protease inhibitors.

## 4. Use of MDSCs in Immunotherapy

Immunotherapy triggers, promotes, or reduces various immune responses; this offers an alternative treatment approach for patients with cancer, inflammatory disorders, and autoimmunity diseases, and for those who require organ transplantation. MDSCs exhibit prominent immunomodulatory features in cancer, inflammation, infection, and organ transplantation [57].

Various therapeutic strategies involving MDSCs are currently being explored. Retinoic acid, a metabolite of vitamin A, facilitates the differentiation of myeloid progenitor cells into DCs and macrophages [105]. SCF induces the expansion of MDSCs in tumor-bearing mice. The suppression of SCF-mediated signaling pathway by blocking the interaction between SCF and its receptor, c-Kit, reduces MDSC proliferation and tumor angiogenesis [106]. Li et al. demonstrated that the adoptive transfer of MDSCs that are regulated by hepatic stellate cells effectively reversed the progression of experimental autoimmune myasthenia gravis, a B-cell-mediated and T-cell-dependent model of myasthenia gravis [105]. Kurko et al. reported that MDSCs produced in vitro markedly suppressed T-cell immunity and arthritis progression in a mouse model of rheumatoid arthritis [99]. The use of granulocyte colony-stimulating factor, which is used to activate hematopoietic stem cells, influences the proportion of MDSCs in the peripheral blood, which, in turn, prevents acute graft-versus-host disease [107].

## 5. Sources of MDSCs

The most commonly used source of MDSCs for immunotherapy is the spleen of tumor-bearing mice. Spleen MDSCs have been widely studied for their immunoinhibitory activities, cell signaling pathways, and differentiation [102]. The second most commonly used source of MDSCs is the blood; a relatively high number of sufficiently pure circulating MDSCs can be obtained from patients or mice with tumors. MDSCs harvested from the blood and spleen differ, functionally and phenotypically, from the tumor-infiltrating subsets. Subsequently, findings that are obtained from using them must be generalized with caution.

Several ex vivo methods have been developed for the propagation of MDSCs. The use of high levels of GM-CSF alone, or GM-CSF in addition to other factors, represents the most common method for the propagation of MDSCs from BM cells or PBMC [108]. Thus, MDSCs can be developed using several ex vivo methods, with varying cytokines and culture conditions. Lechner et al. used a total of 15 mixed cytokines to produce MDSCs from the PBMC of healthy people in vitro, and evaluated their ability to suppress the proliferation of fresh autologous human T cells stimulated by CD3/CD28 and the production of IFN-γ from these cells [109]. However, most of the aforementioned methods resulted in low MDSC differentiation, and a maximum of approximately 50% of MDSCs.

## 6. Characteristics of MDSCs in DKD

### 6.1. The Distribution of MDSCs in DKD

Our previous research demonstrated that the populations of MDSCs in STZ-treated mice were lower in the BM (46.9% vs. 63.2%) than in untreated mice, whereas the proportion of MDSCs increased in the peripheral blood (36.5% vs. 20.5%), spleen (6.8% vs. 3.93%), and kidneys (0.304% vs. 0.225%). The dysregulation of glomerular cells, including endothelial cells, mesangial cells, and podocytes, produces more ROS (Figure 1); furthermore, more pro-inflammatory cytokines are secreted [110] in a diabetic environment to create an inflammatory state that triggers the redistribution of MDSCs from bone marrow to peripheral organs, including the peripheral blood, spleen, and kidneys. The production of MDSCs was significantly reduced when cocultured with mouse mesangial cells (MMCs) under hyperglycemic conditions, from in vitro coculture [110]. This result demonstrated that MMCs probably attenuate MDSC development under hyperglycemic conditions (Figure 2). Islam et al. also presented similar findings that the ratio of MDSCs in type 2 DKD patients is higher than in healthy individuals (6.7% vs. 2.5%) in the peripheral blood. Polymorphonuclear (PMN)-MDSCs accounted for 96% of MDSCs, and the expansion of PMN-MDSCs was not related to the stage of type 2 DKD [111].

### 6.2. The Functions of MDSCs in DKD

It is well known that arginase I and inducible NO synthase (iNOS) participate in the immunoregulatory activities of MDSCs. In our study, MMC-directed MDSCs in high glucose levels expressed lower levels of arginase 1 and iNOS than in normal glucose levels. Under high glucose conditions, MDSCs mediated by MMCs significantly reduced their immunosuppressive function. MDSCs expressed less costimulatory CD80, CD86, and MHC Class II, and more inhibitory B7H1, under the influence of MMCs in the normal glucose environment, indicating that MMC-directed MDSCs represent further immunosuppressive activity than do MDSCs alone. On the contrary, MMC-directed MDSCs weaken the immunosuppressive function of these cells in a hyperglycemic environment [110]. Islam et al. demonstrated that the production of ROS in PMN-MDSCs of type 2 DKD patients is higher than in neutrophils of patients or in the immune cells of healthy individuals, and that this production is enhanced under hyperglycemic conditions [111].

### 6.3. MDSCs Influence T Cell Activities under Hyperglycemic Conditions

MDSCs conditioned by MMC induced more allogeneic T cell activation and less regulatory T cell differentiation in the hyperglycemic state than at normal glucose levels. More interferon-gamma (IFNγ), a pro-inflammatory cytokine, was secreted from T cells that were stimulated by MMC-conditioned MDSCs under hyperglycemic conditions, compared with MMC-conditioned MDSCs in a normal glucose environment. In summary, MDSCs cocultured with MMCs under hyperglycemic conditions weaken inhibitory immune activities, and create an inflammatory environment [110].

## 7. Immunotherapy with Cytokine-Induced MDSCs Improves DKD in Diabetic Mice

The immunomodulatory effects of MDSCs, induced by a combination of GM-CSF, IL-1β, and IL-6, are stronger than those of MDSCs induced by GM-CSF alone, GM-CSF and IL-1β, or GM-CSF and IL-6. MMCs exhibit decreased production of fibronectin mRNA in the presence of cytokine-induced MDSCs (cMDSCs) that are induced by a combination of GM-CSF, IL-1β, and IL-6, particularly under hyperglycemic conditions. The adoptive transfer of cMDSCs into streptozotocin-induced diabetic mice does not alter blood glucose levels or body weight. Nevertheless, the administration of cMDSCs reduces kidney to body weight ratio, fibronectin expression, and fibronectin accumulation in renal glomeruli in diabetic mice. These results indicate that cMDSCs exhibit anti-inflammatory activities that help to improve DKD [110].

Further studies are warranted to elucidate the exact signaling pathways via which MDSCs influence the protein expression of renal mesangial cells under hyperglycemic conditions, and the mechanisms underlying the cross talk between MDSCs and various glomerular cells, such as endothelial cells and podocytes.

## 8. Conclusions and Future Perspectives

Conventional therapy for DKD includes intensive treatment with renin-angiotensin system blockers and sodium-glucose cotransporter-2 inhibitors, and the management of glucose levels and blood pressure [112,113]; however, the prevalence of DKD remains high. Unlike traditional therapies, adoptive cell therapy with MDSCs represents a promising therapeutic strategy for DKD. The administration of cytokine-induced MDSCs to diabetic mice may normalize GFR, decrease kidney to body weight ratio, and reduce fibronectin accumulation in renal glomeruli, thus ameliorating DKD. In conclusion, MDSCs exhibit the anti-inflammatory activities that help to improve renal fibrosis in diabetic mice. The therapeutic targeting of the proliferative or immunomodulatory pathways of MDSCs may represent an alternative immunotherapeutic strategy for DKD.

## Figures and Tables

**Figure 1 ijms-23-13263-f001:**
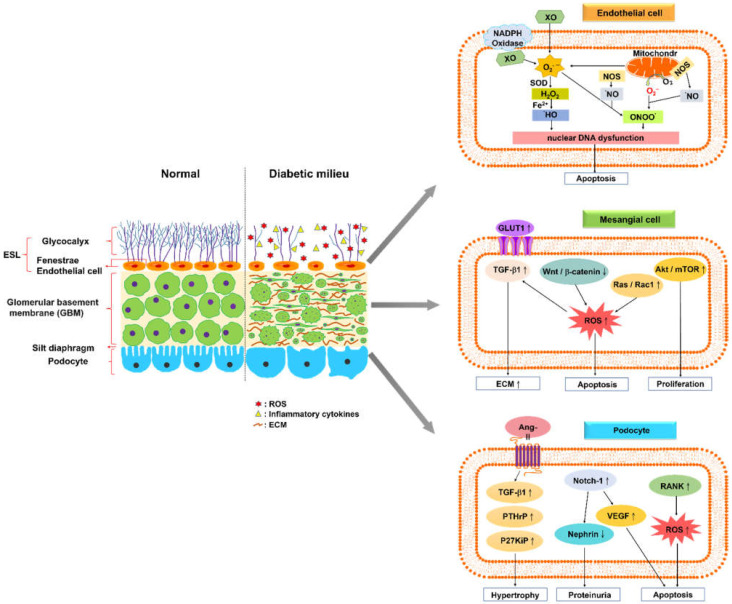
Normal glomerular filtration barrier and its alteration in diabetic kidney disease. A schematic of the normal glomerular filtration barrier, comprising the endothelial surface layer (ESL), glomerular basement membrane (GBM), and podocyte foot processes with slit diaphragm. The glomerular endothelium and fenestrae are covered by the glycocalyx, a meshwork of membrane-bound glycoproteins and proteoglycans. A diabetic kidney is characterized by a fenestrated area of endothelial cells with a reduced amount of glycocalyx, increased levels of reactive oxygen species and inflammatory cytokines, and altered interaction between glomerular endothelial cells and neighboring glomerular cells. Glomerular endothelial cells (GEC) generate reactive oxygen species (ROS), such as superoxide (O_2_^−^) or hydroxyl (•OH) and hydrogen peroxide (H_2_O_2_), via several endogenous pathways, including oxidative phosphorylation in mitochondria, NADPH oxidases, xanthine oxidase (XO), and uncoupled endothelial nitric oxide synthase (NOS). These ROS lead to GEC apoptosis, resulting from nuclear DNA dysfunction. High levels of extracellular glucose increase its uptake through the overexpression of glucose transporter 1, and stimulate transforming growth factor (TGF)-β1 signaling pathways in mesangial cells to induce the production of extracellular matrix proteins. The formation of oxidative radicals may inhibit Wnt/β-catenin signaling or stimulate Ras/Rac1 signaling, which induces the apoptosis of mesangial cells. The Akt/mTOR-mediated autophagy signaling pathway leads to mesangial cell proliferation and fibrosis. Angiopoietin-2–TGF-β1 signaling enhances the aggravation of podocyte hypertrophy under hyperglycemic conditions. The activation of Notch-1 signaling in podocytes causes vascular endothelial growth factor overexpression, nephrin underexpression, and apoptosis augmentation. The overexpression of the receptor activator of NF-κB in podocytes under hyperglycemic conditions increases glomerular oxidative stress and pro-inflammatory cytokine production (↑: up-regulation, ↓: down-regulation).

**Figure 2 ijms-23-13263-f002:**
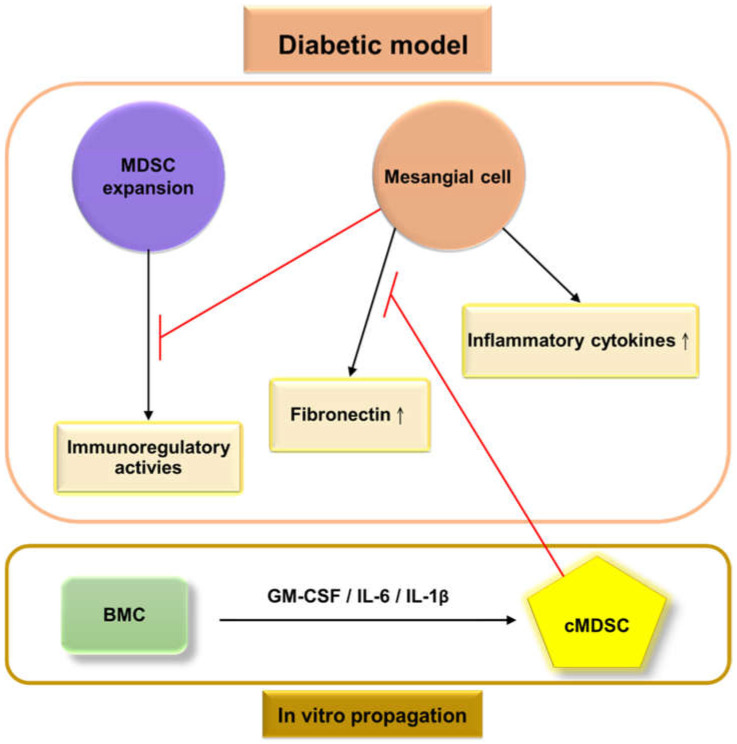
Frameworks for the interactions between glomerular cells and myeloid-derived suppressor cells (MDSCs) under the
diabetic model. High levels of extracellular glucose stimulate mesangial cells to produce high levels of extracellular matrix proteins (primarily fibronectin) and pro-inflammatory cytokines, which then contribute to glomerulosclerosis, and create an inflammatory environment. The bone marrow allocates MDSC cells to peripheral organs. Mesangial cells weaken the proliferation and immunosuppressive effects of MDSCs under hyperglycemic conditions. The
administration of cytokine-induced MDSCs (cMDSCs), propagated from bone marrow cells (BMC) by a combination of granulocyte-macrophage colony-stimulating factor, interleukin (IL)-6, and IL-1β, leads to reduced fibronectin production from mesangial cells, thus improving renal fibrosis; 
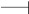
: inhibition.

**Table 1 ijms-23-13263-t001:** Factors involved in MDSC propagation.

Tumor-Derived Factors	Type of Cancer or Tissue Cells	MDSC Population	Refs
GM-CSF	Melanoma (human)	+	[59]
VEGF	Mammary carcinoma (mouse)	+	[60]
IL-1β	Mammary carcinoma (mouse)	+	[61]
IL-6	Fibrosarcoma (mouse)	+	[62]
	Hepatic carcinoma (mouse)	+	[63]
	Prostate carcinoma (mouse)	+	[64]
	Esophageal carcinoma (human)	+	[65]
TGF-β	Mammary carcinoma (mouse)	+	[66]
IFN-γ	Lymphoma (mouse)	+	[67]
PGE2	Ovarian carcinoma (human)	+	[68]
	Lung carcinoma (mouse)	+	[69]
	Melanoma (human)	+	[70]
Hsp72	Lung carcinoma (human)	+	[71]
	Colon carcinoma (mouse)	+	
Flt3 ligand	Mammary carcinoma (mouse)	+	[72]
C5a	Lung carcinoma (mouse)	+	[58]
ATRA	Cervical carcinoma (human)	−	[73]
	Renal carcinoma (human)	−	[74]
S100A9 and S100A8	Myelodysplastic syndromes (mouse)	+	[75]
	Mammary carcinoma (mouse)	+	[76]
SCF	Colon carcinoma (mouse)	+	[77]
**Growth factors and cytokines**			
GM-CSF	ND	+	[78]
G-CSF	ND	+	[79]
IL-6	Hepatic stellate cells	+	[80]
Poly(I:C)	ND	+	[81]
C3	Hepatic stellate cells	+	[82]

GM-CSF, granulocyte-macrophage colony-stimulating factor; VEGF, vascular endothelial growth factor; IL-1β, interleukin-1β; IL-6, interleukin-6; TGF-β, transforming growth factor-β; IFN-γ, interferon-γ; PGE2, prostaglandin E2; Hsp72, heat shock protein 72; Flt3 ligand, fms-like tyrosine kinase 3 ligand; C5a, complement component 5a; ATRA, all-trans-retinoic acid; SCF, stem cell factor; G-CSF, granulocyte-stimulating factor; poly(I:C), polyinosinic:polycytidylic acid; C3, complement component 3; ND, not determined; **+**, increased; and **−**, decreased.

## Data Availability

Not applicable.

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
