# Peer review of "Immune Modulation by Myeloid-Derived Suppressor Cells in Diabetic Kidney Disease"

_ijms, 2022, doi:10.3390/ijms232113263_

Round 1

Reviewer 1 Report

The authors aimed to review an interesting subject. Since the work is a significant contribution to the field, well organized and comprehensively described, it could be published after moderate revisions;

1- Although the origins of MDSCs id well described, and their characteristics are listed in Table 1, more information about the alteration of this population in diabetic kidney diseases is needed. I would like to suggest adding a subheading or possibly a table describing this alteration in different conditions of DKD. 

2-More information about the means of MDSCs recruitment during DKD is needed. 

3- How different clinical conditions associated with diabetes (such as hyperglycemia) may affect MDSC function? Another subheading is needed. 

4- All subheadings related to 3.4. (Mechanisms underlying the suppressive effects of MDSCs) should be discussed more. You may add enough literature to 3.4.1 to 3.4.4 subheadings. 

5- The interaction of MDSCs with other immune cells during DKD is not well-established. 

Reviewer 2 Report

The Review article entitled “Immune Modulation by Myeloid-Derived Suppressor Cells in Diabetic Kidney Disease” aims to highlight how myeloid-derived suppressor cells (MDSCs) may be used as another immunotherapeutic approach to treat diabetic kidney disease (DKD). While this Review seems to touch an interesting topic, the article talks more about MDSCs alone, not even in the context of DKD. Thus, it remains elusive whether using of MDSCs might be a novel promising approach to treat DKD and what are the underlying mechanisms are involved. While other sections make some sense, it is not clear to me what is the goal to place sections 3.2-3.4. It all has information which is not related to DKD. Moreover, unexpected adding of information on lupus nephritis (section 7) is really confusing. Overall, I think this Review needs to be re-organized to archive the goal to discuss how MDSCs may help to treat DKD.

Some minor critique is below:

-        Figure 1: increase font; add endothelial cells or explain in the text why endothelial cells are not involved;

-        Table 1: make in bold “Growth factor and cytokines” title

-        Figure 2: why the thickness of the arrow is different? Does it mean that some molecules/cells contribute more than others? cMDSC is not in the figure legend, please add a description on what is it.

-        Please provide a Figure on how MDSCs improves DKD. Section 8 seems to rely on only one publication. Is there other data that can be added to support the idea that using of MDSCs is good to protect from DKD?

Round 2

Reviewer 2 Report

The manuscript has been significantly improved since the first revision.

Figure 1 still needs some work on increasing fonts inside the blocks as it is difficult to read in the present state.
